# Comprehensive Evaluation of a 1021-Gene Panel in FFPE and Liquid Biopsy for Analytical and Clinical Use

**DOI:** 10.3390/ijms26135930

**Published:** 2025-06-20

**Authors:** Angeliki Meintani, Mustafa Ozdogan, Nikolaos Touroutoglou, Konstantinos Papazisis, Ioannis Boukovinas, Cemil Bilir, Stylianos Giassas, Tansan Sualp, Sahin Lacin, Jinga Dan Corneliu, Paraskevas Kosmidis, Tahsin Ozatli, Dimitrios Ziogas, Maria Theochari, Konstantinos Botsolis, George Kapetsis, Aikaterini Tsantikidi, Chrysiida Florou-Chatzigiannidou, Styliani Maxouri, Vasiliki Metaxa-Mariatou, Dimitrios Grigoriadis, Athanasios Papathanasiou, Georgios N. Tsaousis, Panagoula Kollia, Ioannis Trougakos, Andreas Agathangelidis, Eirini Papadopoulou, George Nasioulas

**Affiliations:** 1Genekor Medical S.A., 15344 Athens, Greece; a.meintani@genekor.com (A.M.); g.kapetsis@genekor.com (G.K.); tsantikidi.k@genekor.com (A.T.); c.chatzigiannidou@genekor.com (C.F.-C.); s.maxouri@genekor.com (S.M.); bmetaxa@genekor.com (V.M.-M.); d.grigoriadis@genekor.com (D.G.); a.papathanasiou@genekor.com (A.P.); gtsaousis@genekor.com (G.N.T.); gnasioulas@genekor.com (G.N.); 2Division of Medical Oncology, Memorial Antalya Hospital, Antalya 07025, Türkiye; ozdoganmd@yahoo.com; 3Department of Medical Oncology, Interbalkan Medical Center, 55535 Thessaloníki, Greece; iat.touroutoglou@gmail.com (N.T.); botsolis.oncologist@gmail.com (K.B.); 43rd Department of Medical Oncology, European Interbalkan Medical Center, 55535 Thessaloniki, Greece; k.papazisis@oncomedicare.com; 5Department of Medical Oncology, Bioclinic Hospital, 54622 Thessaloniki, Greece; ibouk@otenet.gr; 6Department of Medical Oncology, Faculty of Medicine, Sakarya University, Sakarya 54187, Türkiye; cebilir@yahoo.com; 7IASO, General Maternity and Gynecology Clinic, 15123 Athens, Greece; sgiassas@yahoo.com; 8Tansan Oncology, Istanbul 34365, Türkiye; tansan95@gmail.com; 9Department of Medical Oncology, Faculty of Medicine, Koc University, Istanbul 34010, Türkiye; sahin.lacin@hotmail.com; 10Neolife Medical Center, 077190 Bucharest, Romania; 11Second Department of Medical Oncology, “Hygeia” Hospital, 15123 Athens, Greece; parkosmi@otenet.gr; 12Istinye University Hospital, Istanbul 34517, Türkiye; tahsin.ozatli@isu.edu.tr; 13General Hospital of Athens “LAIKO”, 11527 Athens, Greece; ziogasdc@gmail.com; 14Ippokrateio General Hospital of Athens, 11527 Athens, Greece; mtheochari@gmail.com; 15Department of Genetics and Biotechnology, Faculty of Biology, School of Physical Sciences, National and Kapodistrian University of Athens, 10679 Athens, Greece; pankollia@biol.uoa.gr (P.K.); agathan@biol.uoa.gr (A.A.); 16Department of Cell Biology and Biophysics, Faculty of Biology, National and Kapodistrian University of Athens, 10679 Athens, Greece; itrougakos@biol.uoa.gr

**Keywords:** molecular profiling, targeted treatment, immunotherapy, microsatellite instability (MSI), tumor mutational burden (TMB), liquid biopsy, biomarkers, next-generation-sequencing (NGS) panel

## Abstract

In the era of precision oncology, comprehensive molecular profiling is critical for guiding targeted and immunotherapy strategies. This study presents the analytical and clinical validation of a 1021-gene next-generation sequencing (NGS) panel, designed for use with both formalin-fixed paraffin-embedded (FFPE) tissue- and liquid-biopsy specimens. Analytical validation confirmed the assay’s high sensitivity and specificity across variant types—including SNVs (Single Nucleotide Variations), indels, CNVs (Copy Number Variations), and fusions—down to a 0.5% variant allele frequency. The assay also accurately identified microsatellite instability (MSI) and tumor mutational burden (TMB), essential biomarkers for immunotherapy. Clinical validation was performed on over 1300 solid tumor samples from diverse histologies, revealing actionable alterations in over 50% of cases. The panel detected on-label treatment biomarkers in 12.57% of patients, increasing to 20.15% when immunotherapy markers were included. Additionally, the assay demonstrated strong concordance with orthogonal methods and was effective in detecting variants in plasma-derived circulating tumor DNA in 70% of evaluable cases. These findings support the robust performance and broad clinical applicability of the 1021-gene panel for comprehensive genomic profiling in both tissue and liquid biopsies, offering a valuable tool for personalized cancer treatment.

## 1. Introduction

Accurate analysis of tumor biomarkers is essential for guiding optimal treatment options for cancer patients based on their individual disease characteristics. The analysis of large series of genes using next-generation sequencing (NGS) technologies is currently the most favorable approach to obtain a comprehensive review of tumor biology in the era of precision medicine [1].

Several approved or investigational drugs targeting specific tumor characteristics and/or dysregulated biological pathways are being studied for their role in modulating treatment sensitivity and resistance. Moreover, in parallel to targeted therapies, immunotherapies are also evolving in the context of cancer therapy [2]; more specifically, immune checkpoint inhibitors are widely used as anticancer immunotherapeutics, having been approved for use in a variety of tumors. However, given their toxicity and high cost, identifying reliable biomarkers for guiding therapeutic decisions remains a critical challenge [3].

To address the growing need for comprehensive tumor analysis, broad NGS gene panels have been developed to provide predictive insights into responses to targeted therapies and immunotherapy. Most importantly, the selection and validation of the NGS methodology should comply with current recommendations for assay performance. The accuracy and sensitivity of the selected methodology strengthen the validity of the results and ensure an appropriate treatment approach for cancer patients [4,5].

In addition, even though the utilization of tumor tissue is still considered the gold standard approach for cancer-related DNA analysis, there are still several limitations, such as unavailability or insufficient quantity/quality of the obtained genetic material. Therefore, the use of liquid biopsies, particularly plasma-derived cell-free DNA, offers a promising approach for obtaining a comprehensive molecular profile of the tumor and capturing treatment-related information that might otherwise be overlooked [6]. Consequently, it is of great importance to make an appropriate selection of genes for NGS analysis and ensure the quality of the results based on the type of genetic material. In this context, the parallel analyses of formalin-fixed paraffin-embedded (FFPE) tissue and plasma samples have been shown to provide complementary information, increasing the actionability of the results. Hence, genomic alterations with possible associations with resistance to treatment can be identified in a more robust way, while a more comprehensive biological overview of primary, as well as metastatic tumors, can be obtained [7].

Based on these considerations, the reliability and applicability of a 1021-gene NGS assay, the Oncology Multi-Gene Variant Assay (GenePlus, Beijing, China), for tumor molecular profiling in both tissue- and liquid-biopsy specimens were investigated. The sensitivity and specificity of the process for detecting genetic variants were assessed to ensure appropriate biomarker analysis in cancer patients, while the reliability of immunotherapy-related biomarkers, including microsatellite instability (MSI) and tumor mutational burden (TMB), was also assessed to provide a comprehensive overview of predictive markers for treatment response.

## 2. Results

### 2.1. Assay Performance Quality Metrics

In order to assure the accuracy of the NGS data, quality control metrics suggested by the manufacturer, including library concentration, average on-target sequencing depth without duplicated reads, fraction of base quality ≥ Quality Score (Q)30, and fraction of target covered with ≥50×, were assessed (Table 1). An analysis of the unique molecular identifiers (UMIs) generated sequencing data volumes of 5 GB (Gigabytes) and 17 GB, which were required to achieve mean coverages of 500× and 2000×, respectively. The proposed thresholds for the analysis of the UMIs were empirically validated and, subsequently, confirmed using the analytical assay validation results.

The assay achieved an average coverage of >500× after analysis of the UMIs for all tested reference and control samples. The mean proportion of bases exhibiting a quality value of ≥Q30 was 94.7%. The correct uniformity was obtained with 99.95% of the bases covered at a quality value of >50×.

### 2.2. Validation Materials

#### 2.2.1. Reference Materials Results

The S800-1 reference sample was used, exhibiting a variant allele frequency (VAF) of 2% for clinically relevant variants in the *EGFR*, *KRAS*, *NRAS*, and *KIT* genes, as well as the *ALK* and *ROS1* gene fusions. The S800-2 reference sample, with a VAF of 0.5%, was also employed. Both standards carried an *ERBB2* gene amplification. These variants were detected due to the appropriate sequencing depths achieved and the appropriate amount of GB generated. The positive percent agreement (PPA) and negative percent agreement (NPA) were 100% for all individual variants (SNVs, indels, fusions, and CNVs), highlighting the robustness and accuracy of our experimental workflow (Table 2).

In addition, The Oncospan reference standard and the Structural Multiplex reference standard were used. OncoSpan is a well-characterized, cell-line-derived reference standard containing 386 variants across 152 key cancer genes. Variants are present between a 1 and 100% allelic frequency (AF), with 52 variants present at ≤20% AF for LOD determination of your assay (catalog ID: HD827). The Structural Multiplex includes 9 ddPCR-validated mutations, with most of them centered at a 5% allelic frequency. Highlight features of the Structural Multiplex include the *RET* and *ROS1* fusion variants, as well as *MYC-N* and *MET* focal amplifications. All variants present in the Oncospan and the Structural Multiplex reference standards were accurately detected.

In order to calculate the lower limit of detection, the evaluation of the assay performance for the SNVs/indels was estimated at 1–1.3% VAF using the Tru-Q 7 (1.3% Tier) reference standard and 0.5–0.65% VAF when using an equimolar mix of Tru-Q 7+ Tru-Q0 (wild type), respectively. In the case of 1–1.3% VAF, a mean 993x coverage led to a 97.50% agreement among the expected and detected variants, which was slightly reduced to 92.5% at a 560× coverage. Concerning the lower VAF variants (0.5–0.65), an 85% agreement was observed at a 760× coverage.

Therefore, the sensitivity for 2%, 1.3%, and 0.6% VAF was 100% (95% CI 0.844958–1), 96.15% (95% CI 0.884086–0.990014), and 84.62% (95% CI 0.687915–0.935864), respectively.

The inter-assay repeatability was evaluated using three independent DNA libraries of the S800-1 and S800-Wild-Type reference standards. Libraries were prepared and sequenced in triplicate on the same day, flow cell lane, and system. In addition, a DNA sample extracted from an FFPE sample was also prepared and sequenced in triplicate. To evaluate the assay reproducibility, libraries from the same FFPE samples were also prepared and sequenced on multiple days, by different operators.

#### 2.2.2. Tissue Cohort Results

The minimum sample input was determined through the analysis of 3 DNA samples with 50, 100, and 200 ng of DNA as starting material. The sequencing data volume ranged from 5.4 to 6.2 GB, and consistent genetic analysis results were obtained with all replicates. In addition, the Oncomine Comprehensive plus (OCA plus) panel (Thermofischer scientific, Waltham, MA, USA) was used for the analysis of 50 tissue samples harboring 131 actionable variants, analyzed in GENEKOR’s laboratory [8]. These samples carried all types of variations, namely, SNVs, indels, gene rearrangements, and gene amplifications. All variants that were detectable by the OCA were also detected by the 1021-gene panel, with the exception of an *NRG* fusion. In addition, an *RAD51C* alteration was detected only by the 1021-panel due to the low coverage of this region in the OCA plus panel (Appendix A). Furthermore, the 1021-gene panel was applied to 12 HER2-positive and 10 *HER2*-negative samples, as determined by Fluorescence In situ Hybridization (FISH). The NGS assay accurately classified all FISH-negative samples as negative, while ten out of twelve FISH-positive samples were also identified as positive by NGS; hence, the combined accordance between the two methodologies amounted to 90.91%.

#### 2.2.3. FFPE and Liquid Biopsy Comparative Analysis Results

Plasma samples were obtained from 10 patients with available FFPE results using the OCA plus assay, to test the applicability of the assay for liquid-biopsy analysis (Table 3). In 7/10 cases, the alterations detected in the tissue samples were also present in the paired plasma samples. In the remaining three cases, tissue alterations were not detected in the respective plasma samples, indicating the absence of circulating tumor DNA. This result was probably because all patients were subjected to treatment at the time of plasma collection.

### 2.3. MSI Results’ Validation

A total of 66 samples with a known MSI status, using either the OCA plus or a validated PCR assay, were used to test the validity of the MSI results. Among these, 49 were characterized as MSI stable and 17 as MSI high. The 1021 assay detected all 17 positive samples (sensitivity 100%) and correctly classified 48 out of 49 MSS samples as stable (specificity: 98%).

### 2.4. TMB Results’ Validation

Fifty-nine FFPE samples previously evaluated for TMB using the OCA plus assay were analyzed using the 1021-gene panel. In addition, 2 TMB control samples were evaluated (30 muts/MB and 10 muts/MB). A high correlation between the expected and observed values was noticed (r(59) = 0.82, *p* = < 0.00001) (Figure 1).

### 2.5. LOH

The LOH values obtained by the NGS and the Oncoscan assay were found to be moderately positively correlated, r(33) = 0.6782, *p* < 0.00001. This indicates that high LOH values by NGS are typically accompanied by high LOH values by Oncoscan and vice versa.

### 2.6. Clinical Validation Results

The clinical validation of the panel was performed in over 1000 patients with various types of solid tumors who were referred for molecular profiling by their treating oncologists. The most prevalent tumor type was pancreatic cancer followed by lung, colorectal, and breast cancers (Figure 2). Of these, 188 cases (12.08%) were excluded from the experimental procedure due to the lack of available cancer tissue material. Among the remaining 1368 cases, successful NGS genomic testing was achieved in 1345 patients (98.32%). Among the patients tested, 645 were male and 700 were female (47.96% and 52.04%, respectively). The average age of patients at diagnosis was 58.36 years across both genders (59.73 years for males and 57.10 years for females).

Genomic analysis in 1301 (96.73%) patients led to the identification of a total of 10,805 genomic alterations in 692 different genes. Of these, 7160 alterations (66.3%) were classified as oncogenic/likely oncogenic based on the joint set of guidelines developed by the Clinical Genome Resource (ClinGen), Cancer Genomics Consortium (CGC), and Variant Interpretation for Cancer Consortium (VICC) [9]. The remaining 3652 variants (33.7%) were categorized as variants of uncertain significance (VUS).

Among the oncogenic variants, the majority (51.56%) were SNVs or small indels (36.19% and 15.38%, respectively), while the remaining variants (47.39%) concerned CNVs and gene fusions (1.05%). The *TP53* gene was the most frequently affected gene, carrying an alteration in 47.66% of the patients followed by the *KRAS*, *CDKN2A*, *PIK3CA,* and *TERT* genes (26.25%, 14.41%, 12.79%, and 11.52%).

Regarding the actionability of the detected oncogenic variants, 12.57% of the patients carried a variant associated with on-label treatment, 43.94% of the cases carried off-label treatment-related biomarkers, while 5.95% of the cases carried variants associated with resistance to treatment. Among the 169 patients with on-label biomarkers, the most prevalent targeted gene was *BRAF* that accounted for 18.34% of the cases. Frequent targeting was also observed in *ERBB2* (14.79%), *PIK3CA* (10.06%), and *BRCA1/2* (8.28%), as well as the *KRAS*, *EGFR,* and *ESR1* genes (5.92% each) (Figure 3).

The actionability of the variants was highly associated with the tumor type. Therefore, variants associated with on-label treatment were detected in 37.27% of breast cancer patients, 25.64% of prostate cancer patients, 24.35% of lung cancer patients, and 21.13% of the patients with biliary tract tumors. On the other hand, tumors such as cervical, endometrial, pancreatic cancer, and sarcoma exhibited low frequencies of variants associated with on-label treatment (0%, 0%, 1.61%, and 0.89%, respectively) (Appendix A, Appendix A).

In addition to on-label biomarkers several off-label biomarkers are also detected in our cohort. The PTEN alterations are particularly common in various tumor types, including endometrial (36.67%), brain (30.47%), cervical (23.81%), thyroid (13.63%), sarcoma (10%), and HNSCC (7.69%), underscoring its broad relevance in tumorigenesis [10]. PIK3CA alterations are also frequently detected across cancer types, including ovarian (17.14%), urothelial (11.54%), and colorectal cancer (11.43%), reflecting its known role in PI3K pathway dysregulation [11]. Moreover, KRAS alterations drive oncogenesis in the majority of pancreatic cancers (73.40%), while a high proportion of biliary tract tumors (25.35%) presents such alterations [12,13]. Given their central role in tumorigenesis across diverse cancer types, such frequently mutated genes represent promising candidates for pan-cancer therapeutic targeting.

Our assessment extended to the evaluation of somatic variants occurring in 26 homologous recombination-repair (HRR) genes, which were included in our NGS panels. A loss-of-function variant was detected in 296 out of 1345 patients (19.78%). The *BRCA1/2* genes carried variants in 3.72% of the analyzed cases, whereas 16.06% of the cases showed alterations in other genes, with *ATM* being the most prevalently targeted gene. Only 5.35% of the patients displayed high-risk gene alterations, such as *BRCA1/2*, *PALB2*, and *RAD51C*. In terms of clinical relevance, most of the HRR variants were found in genes categorized as intermediate- or low-risk, such as *ATM*, *CHEK2*, *FANCA*, *BAP1*, and *FANCM*, among others, all of which have shown uncertain predictive significance (Figure 4). Of note, 16.67% of the analyzed cases harbored more than 1 HRR gene alteration, indicating a possible role of this pathway in the oncogenesis process.

In addition, high gLOH was calculated for the 70 ovarian cancer patients. In agreement with previous reports (10), elevated gLOH was observed in 51.43% of the cases. Combining gLOH and alterations in high-risk HRR genes, 57.14% of ovarian cancer patients would be eligible for Poly (ADP-ribose) polymerase (PARP) inhibitors treatment.

Immunotherapeutic biomarkers, such as MSI and TMB, were also assessed in all cases. Only 16 patients (1.19%) were confirmed to have an MSI-high status, half of whom were colorectal cancer cases. A TMB value of >10 muts/MB was detected in 124 cases (9.22%), of which 25 were lung cancer cases, 21 colorectal, 15 breast, and 8 gastric malignancies. The rest of the cases were mainly sarcomas and urothelial and brain tumors.

Based on the NGS results, the combined investigation for the identification of biomarkers related to targeted therapy and immunotherapy increased the number of patients with biomarkers associated with on-label treatment by almost 8%, reaching 20.15% (Figure 5).

## 3. Discussion

The evolving role of genomics analysis in clinical decision making is unquestionable. Especially in the case of solid tumors, molecular profiling is nowadays complementing histopathological findings for better comprehension of the tumor biological background, leading to differential prognosis and response to treatment, even for histologically identical tumors. Appropriate NGS methodologies for tumor analysis provide actionable information and assist in clinical management and treatment decision making. Presently, NGS analysis of tumors is being widely implemented in clinical practice, particularly for patients with limited treatment options, [1]. Consequently, molecular testing facilities need to employ the most reliable and informative technologies in order to ensure the extraction of robust and meaningful results.

In the present study, we evaluated a multi-biomarker analysis NGS assay for its efficiency in detecting actionable variants related to targeted treatment and immunotherapy biomarkers. The Oncology Multi-Gene Variant Assay (GenePlus) is a qualitative in vitro diagnostic test that detects variants in 1021 tumor-related genes and gene rearrangements/fusions in 38 genes. The analysis of reference FFPE and DNA revealed its efficacy in detecting all types of alterations with high sensitivity and specificity. Therefore, it was able to detect >99% of variants evaluated at a VAF of 2%, with a mean depth of coverage >500×, making it compatible with tumor tissue analysis (Table 1). Reference samples were utilized and confirmed the repeatability and reproducibility of the results obtained. Moreover, the assay was also able to detect critical biomarkers at 0.5% VAF with 100% sensitivity and specificity at a 1000× mean sequencing depth, indicating its compatibility with ctDNA analysis (Table 2 and Table 3).

The concurrent evaluation of numerous tumor biomarkers within a suitable timeframe for clinical decision making and at a cost that is affordable for the patients has been made possible by developments in sequencing technologies and the throughput of NGS platforms. Treatment options may include targeted therapeutics, immunotherapies, or conventional procedures in cases where genetic findings are negative [14]. The efficacy of personalized medicine in selecting treatments has been firmly proven, with several studies demonstrating that a biomarker-guided targeted-therapy approach provides a greater survival benefit compared to conventional cytotoxic treatment [14,15,16,17,18]. Hence, the guidelines of the National Comprehensive Cancer Network (NCCN) delineate the extent to which molecular profile analysis enhances patient care for different types of tumors. In addition, the number of molecularly targeted therapies continues to rise, as they are evaluated in ongoing clinical trials for their safety and efficacy in patients with a variety of advanced solid and hematological malignancies. Ongoing endeavors are made to identify additional genomic or immune-mediated biomarkers that can forecast response to specific treatments, beyond tumor histology, starting with the site-agnostic indication of pembrolizumab for patients with MSI-unstable tumors and the inhibition of TRK in NTRK fusion-positive cancers [19]. At present, seven agnostic biomarkers have received pan-cancer approval, while others, including PARP inhibitors, have demonstrated efficacy across a range of tumor types [20]. The simultaneous analysis of the increasing number of genetic alterations that determine response to treatment is crucial and provides an immediate depiction of the tumor biology that could be utilized for the personalization of a patient’s management.

In a clinical NGS test, the bioinformatics pipeline—primary, secondary, and tertiary analyses—must be evaluated [21]. This ensures appropriate secondary analysis of the raw sequencing data, which is fundamental and requires the utilization of bioinformatic tools for mapping, alignment, and variant calling. During this step, it is important to ensure that the method utilized can accurately detect all types of variants, including SNVs, indels, CNVs, and fusions. Tertiary analysis, which concludes the NGS pipeline, is the juncture at which the implications of each genomic variant detected in the preceding phases for the cancer patient become informative. It includes the annotation of variants followed by the filtering, prioritization, and classification of variants according to their actionability and clinical indications.

Moreover, in the clinical setting the accurate detection of actionable alterations is of limited significance unless it is accompanied by the appropriate interpretation and reporting of the detected variants. This encompasses thorough and documented information of their correlation with approved on-label, off-label, and investigational drug sensitivity or resistance. Effective documentation should integrate relevant guidelines, clinical trials, and literature resources. The combination of the NGS assay’s technical accuracy and accurate clinical interpretation of the molecular findings makes possible the translation of genomic data into personalized therapeutic strategies offering individualized patient care solutions to cancer patients and physicians through this process.

The performance of the assay on the FFPE samples previously analyzed by alternative standard methodologies was also evaluated. It demonstrated significantly concordant results for all biomarkers compared with alternative NGS assays. In addition, MSI, HER2, and ALK fusion results were highly concordant with those of the orthogonal methodologies (PCR and FISH). The TMB results were also in concordance with the alternative NGS methodology (Oncomine Comprehensive plus-OCA plus; Thermo Fisher Scientific, Waltham, MA, USA), while the TMB reference standards were accurately measured.

In our validation study, a direct comparison between the results obtained using the hybrid capture-based 1021-gene panel and the amplicon-based OCA plus assay was performed in 50 clinical samples [8,22]. The results show key performance differences between the assays. The hybrid capture design enabled broader and more consistent interrogation of the genomic regions of the 1021 genes, including difficult-to-amplify or GC-rich loci. The new assay also achieved superior coverage uniformity, aided by the incorporation of unique molecular identifiers (UMIs) for error correction and accurate low-frequency variant detection, which makes it eligible for liquid-biopsy analysis in addition to FFPE tissue testing [23]. It is important to note that the OCA plus assay failed to detect a clinically relevant RAD51C variant in the genomic region due to insufficient coverage, which highlights a common limitation of amplicon-based approaches. These findings are in line with the results of other studies that compared hybrid capture and amplicon-based panels, which demonstrated that hybrid capture was more effective in detecting complex variants and fusions [24,25]. Nevertheless, a notable advantage of the OCA plus assay and similar amplicon-based assays is their ability to operate with lower-input DNA quantities, making them suitable for small or degraded samples. Moreover, while the OCA plus assay offers practical advantages—such as low DNA input and rapid turnaround—it has limited sensitivity for subtle copy number alterations, particularly deletions [26]. In fact, a high concordance of 90.91%.was observed for ERBB2 amplification detection when comparing our hybrid capture methodology with the orthogonal assay (FISH). Overall, while both methods have clinical utility, our results support the broader diagnostic reach and technical robustness of hybrid capture-based profiling, particularly in comprehensive genomic applications.

The 1021-gene panel employed in this study showed comparable clinical utility to that of other hybrid capture-based assays, while its relevance in daily practice is further enhanced by the large panel design and the number of HRR genes that were interrogated [24,27,28,29,30,31,32]. Additionally, even in the absence of an RNA-based analysis, the DNA-only sequencing methodology we used was able to accurately identify gene fusions. This method was dependent on optimal probe design and extensive coverage to identify rearrangements in clinically pertinent fusion partners. DNA and RNA have been employed in several CGP assays to enhance the detection sensitivity of fusions, particularly for genes such as *ALK*, *RET*, *ROS1*, and *NTRK1/2/3* [33]. However, RNA sequencing also has substantial practical and technical pitfalls, particularly when working with formalin-fixed, paraffin-embedded (FFPE) samples, where RNA integrity is frequently compromised. Therefore, although RNA-based methods can directly validate transcript expression and additionally identify rare or novel fusions, they require dual-extraction procedures and high-quality input materials, leading to increased tissue waste, turn-around time, and complexity. Our findings support the clinical value of DNA-based, high-depth sequencing as a means for confident fusion detection, especially in the general diagnostic context where the quality and/or quantity of RNA is often low [32,34].

The NGS methodology utilized provided results in almost every analyzed case, with >20% TCC measuring at least 3 mm × 3 mm of tissue sample available. At least one alteration was identified in the majority of tumor specimens analyzed. Findings related to on-label treatment biomarkers were detected in 12.57% of the patients and this percentage increased to 20.15% in the case of concurrent evaluation of both targeted treatment and immunotherapy biomarkers. In total, over half of the patients harbored a somatic variant related to either targeted treatment or immunotherapy on- or off-label.

The liquid-biopsy NGS analysis, while important for biomarker identification, still presents several challenges. The main limitation is the lower sensitivity in early-stage cancer and in patients undergoing cytotoxic or targeted therapy, since ctDNA levels may be extremely low, leading to false-negative results. Therapy can also suppress ctDNA shedding, rendering it difficult to assess the tumor mutational burden or minimal residual disease precisely during treatment. Furthermore, the detection of some genomic changes, such as gene fusions or copy number variations, remains technically challenging with ctDNA technologies, though when used alongside tissue testing it offers a more comprehensive molecular profile. [35]. However, in the present study, 10 cases with available tissue molecular profile results underwent plasma NGS analysis. In 70% of these cases, the tissue alterations were also detectable in liquid biopsy, indicating the importance of utilizing this NGS procedure, despite its limitations. The findings identified in these cases were point mutations, CNVs, and fusions, highlighting the applicability of this 1021-gene panel in liquid biopsies, as well. The acquisition of negative ctDNA findings in three cases may have been influenced by the fact that patients were undergoing treatment. Thus, the minimal quantity of ctDNA detected necessitates a subsequent analysis following treatment completion in order to ascertain whether the amount of ctDNA detected has increased. Moreover, the applicability of this assay in ctDNA analysis enhances its clinical utility, particularly in cases where tissue biopsy is not available. While the two methods are complementary in providing a comprehensive understanding of tumor biology, liquid biopsy can also be used for real-time monitoring of resistance mutations emerging after targeted therapy. Liquid-biopsy molecular profiling has the potential to increase the number of patients eligible for targeted therapy; therefore, it is recommended by international guidelines (NCCN and ESMO) for a patient’s appropriate management [23].

In the present study, a multigene NGS assay has been extensively validated for its applicability in tissue- and liquid-biopsy molecular profiling. Our results have shown that this assay has excellent performance for actionable alterations analysis while providing accurate results for molecular signatures related to immunotherapy such as TMB and MSI.

## 4. Material and Methods

### 4.1. Analytical Validation

The validation included the nucleic-acid-extraction method, sequencing process, and bioinformatics pipeline. In addition, the assay’s sensitivity, specificity, repeatability, and reproducibility were evaluated using reference samples, as well as clinical samples evaluated by an orthogonal method (Table 4). Custom reference standards with 2% and 0.5% allelic frequencies (catalogue numbers: CA0564 and CA1425, respectively) were utilized for the assay sensitivity, specificity, and reproducibility analyses (GeneWell Biotechnology Co., Ltd., Shenzhen, Guangdong, China). In terms of the reference samples, the Tru-Q 7 (1.3% Tier) and Tru-Q 0 (100% Wildtype) reference standards, as well as the Oncospan (HD832) and the Structural Multiplex reference standards (HD753) (Horizon Discovery Ltd., Cambridge, UK) were used [36,37].

### 4.2. Patients

A total of 1558 patients with various histological tumor types were referred to our laboratory by their oncologists for tumor molecular profiling between November 2023 and December 2024. Most cases involved pancreatic, colon, lung, breast, sarcomas, gastric, and biliary tract malignancies, as well as ovarian, prostatic, and unknown primary cases. Consent to participate in the study was obtained in writing from all patients; pathology reports were available for all cases and were utilized to confirm the clinical diagnosis and select the most suitable tissue specimen for analysis.

### 4.3. Tissue Selection and Nucleic Acid Isolation

Formalin-fixed and paraffin-embedded (FFPE) tumor biopsies were obtained from a tumor area with tumor cell content (TCC) exceeding 20%, measuring at least 3 mm × 3 mm. The procedure was based on instructions by qualified pathologists in sections stained with hematoxylin and eosin. Subsequently, genomic DNA isolation was performed using the MagMAX™ Total Nucleic Acid Isolation Kit (Thermo Fisher Scientific, Waltham, MA, USA). In addition, the ctDNA analysis was performed using plasma-extracted cfDNA. DNA was also extracted from leukocytes to be used as a control to prevent the detection of false-positive results due to clonal hematopoiesis mutations. The MagMAX Cell-Free DNA Isolation Kit (Thermo Fisher Scientific, Waltham, MA, USA) and the MagCore Genomic DNA Whole Blood Kit (RBC Bioscience Co., Ltd., New Taipei City, Taiwan) were used for the cfDNA and genomic DNA extractions, respectively.

### 4.4. NGS Procedure

Targeted next-generation sequencing (NGS) analysis was performed using the Oncology Multi-Gene Variant Assay (GenePlus, Beijing, China), which is a qualitative in vitro diagnostic test capable of detecting variants in 1021 tumor-related genes, as well as gene rearrangements/fusions in 38 genes. Sequencing was carried out on an MGI sequencing platform, DNBSEQ-G400 (MGI Tech Co., Ltd., Shenzhen, China). The analysis included the entire exon regions of 312 genes, introns/promoters/fusion breakpoint regions of 38 genes, and selected/partial coding exons of 709 genes (Appendix A). The test also reports 30+ immune-response biomarkers, including TMB score and MSI status. The test can be applied for the analysis of both FFPE tissue and liquid biopsies of plasma-extracted cfDNA. The ctDNA analysis was performed using plasma-extracted cfDNA in combination with DNA extracted from blood cells as a control to avoid the detection of false-positive results due to clonal hematopoiesis mutations.

Libraries containing cancer-related gene enrichment and sequencing on MGI sequencing platforms (DNBSEQ-G400) were constructed as follows:

**Library Preparation Prior to Capture**: The genomic DNA (gDNA) extracted from FFPE tissue or leukocytes (regarded as the control) was sheared into small fragments and then gDNA fragments were purified with magnetic beads. For the cfDNA extracted from plasma (regarded as the case), the steps involving fragmentation and purification were not necessary. End repair and A-tailing were performed for the selected gDNA fragments. Universal adaptors with known sequences were ligated onto the gDNA fragments. The P5 and P7 index sequences were incorporated into each library at the ends of the repaired gDNA fragments. The indexes include a unique sequence for identifying each individual sample. The pre-capture libraries were obtained after further purification.**Target Enrichment:** Pre-capture libraries were enriched for specific genes of interest using a hybridization capture-based method. Specially designed, biotinylated probes spanning the gene regions of interest were hybridized with the libraries. After hybridization and incubation, the probes and hybridized targeted libraries were isolated from non-targeted libraries using streptavidin magnetic beads. Enriched, targeted libraries were obtained after the application of the washing, amplification, and purification steps.**Library Circularization:** Enriched libraries were denatured into single-stranded DNA at high temperature and circularized into single-stranded circular DNA.**Sequencing:** The single-stranded DNA nanoball (DNB) was prepared for sequencing by using the original single-stranded circular DNA as a template for isothermal amplification (called rolling circle amplification (RCA)). DNA nanoballs were pumped by the fluidics system, loaded onto a patterned array chip (sequencing flow cell) and sequenced using cPAS (combinatorial probe anchor synthesis)-based sequencing on an MGI sequencing platform (DNBSEQ-G400).

### 4.5. Data Analysis and Result Interpretation

The data obtained by NGS were analyzed with a relevant analytical bioinformatics pipeline, while the results’ interpretation was performed with the instrument Gene Box platform (GenePlus, Beijing, China). Single-nucleotide variants (SNVs), insertions, deletions (indels), and structural variants (SVs) were identified as positive in the case of a frequency of ≥1% for both the FFPE and plasma samples. Positivity for the copy number variations (CNVs) required a copy number ≥ 3.0 for the FFPE and 2.2 for the plasma samples, respectively. The relevant cutoff values for MSI-H were ≥19.5% and 18%, respectively.

The tissue TMB calculation was performed as previously reported [38]. Briefly, the TMB algorithm considers all somatic mutations, including synonymous ones; coding area > 1 MB; and SNV/indel, including the splice (±2). For the qualified control samples included in the TMB calculation, only mutations of VAF > 5% were considered, while risk samples were not set on the basis of the variation range reported. Driver mutations, including tumor-suppressor-gene inactivation mutations, driver-gene-sensitive mutations, and driver-gene high-frequency mutations were excluded.

### 4.6. LOH

Our assay measures genomic instability using sample-level LOH in addition to the analysis of 26 alterations in genes related to homologous recombination (HR) (Appendix A). The algorithm utilizes heterozygous population single-nucleotide polymorphisms (SNPs) to determine the ploidy levels of genomic segments. The genome is then divided into contiguous segments of similar ploidy levels. Log odds ratios for the variant allele frequency of the observed population of SNPs and the copy number (CN) ratios for each segment were calculated. Log odds ratio and CN ratios were then used to infer tumor cellularity (i.e., percentage of the tumor cells in the sample) and loss-of-heterozygosity (LOH) for each genomic segment. Segment-level LOH events were intersected with targeted gene boundaries to determine LOH events in selected genes. Segment-level LOH events were also aggregated to determine the sample level % LOH.

### 4.7. OncoScan CNV Assay

The OncoScan™ CNV Assay (Thermo Fisher Scientific, Waltham, MA, USA) was carried out as previously described [39]. The Chromosome Analysis Suite (ChAS) was used for the primary analysis of the output files (in .CEL format) and quality control calculations (MAPD and ndSNPQC). The ASCAT (v3.0.0) (allele-specific copy number analysis of tumors) was used to evaluate and calculate tumor purity, ploidy, and allele-specific copy number profiles, using the logR ratio and B-allele frequency of autosomal markers with GC content, as well as replication timing correction [40,41]. Segmentation data from ASCAT, along with the previously described algorithms and definitions, were used to calculate the % LOH [42,43].

## Figures and Tables

**Figure 1 ijms-26-05930-f001:**
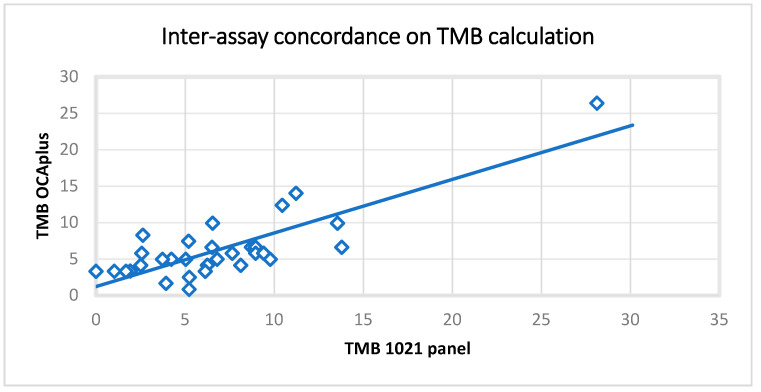
Inter-assay concordance on TMB calculation.

**Figure 2 ijms-26-05930-f002:**
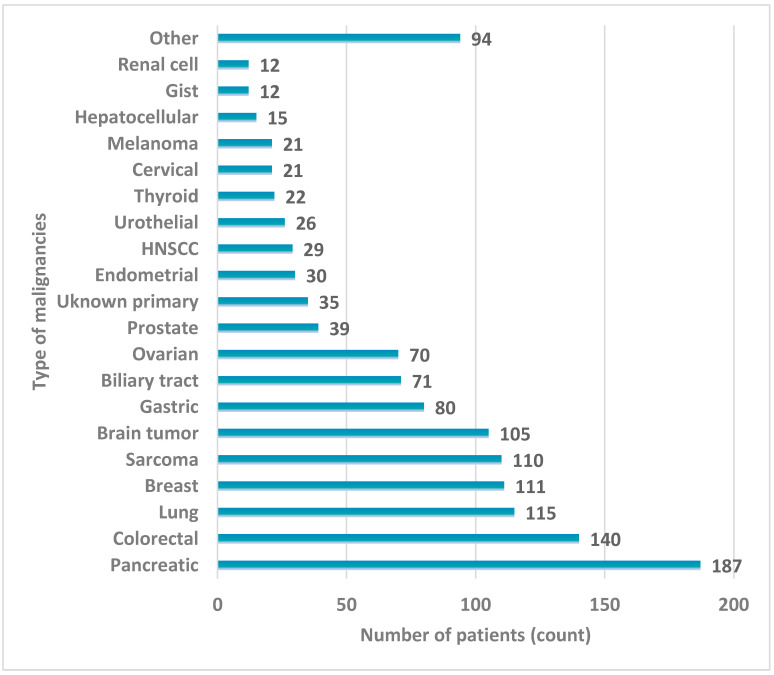
Tumor types included in clinical validation of the 1021-gene assay.

**Figure 3 ijms-26-05930-f003:**
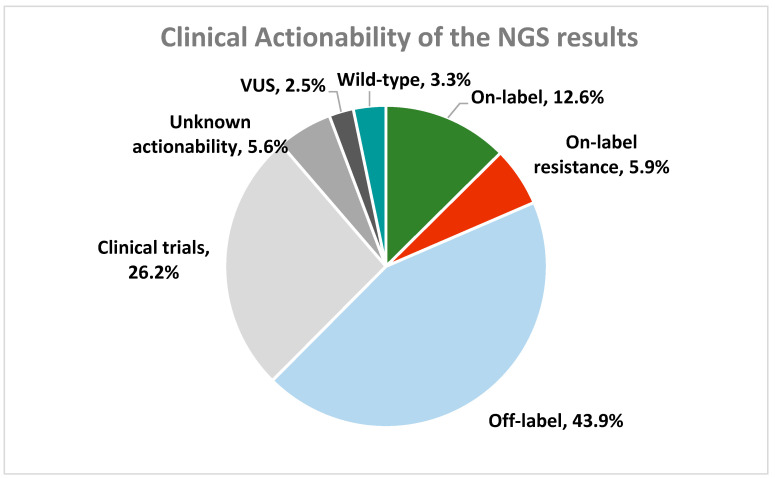
Patients’ categorization based on the most actionable variant detected in all tumors analyzed.

**Figure 4 ijms-26-05930-f004:**
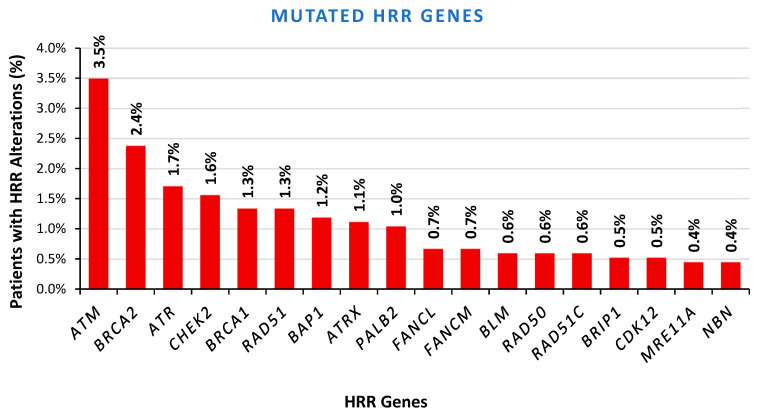
HRR mutation distribution.

**Figure 5 ijms-26-05930-f005:**
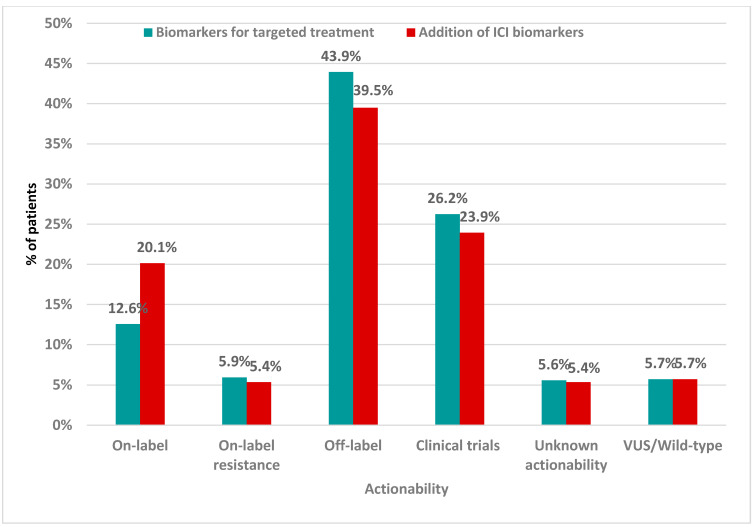
Clustered bar chart showing patient’s categorization based on the most actionable biomarker identified, including both biomarkers for targeted therapy and immunotherapy.

**Table 1 ijms-26-05930-t001:** Quality control metrics.

Quality Control Index		Criterion
DNA Quality Assessment	DNA amount (ng)	≥50
DNA Library Quality Assessment	DNA library amount (ng)	≥600
Sequencing Quality Assessment	Average effective sequencing depth	≥500 (for 2% VAF)/1000 (for 0.5% VAF)
	Fraction of target covered with ≥50×	≥99%
	Fraction of base quality ≥Q30	≥80%

Abbreviations: ng: nanograms. VAF: Variant Allele Frequency.

**Table 2 ijms-26-05930-t002:** Reference control results.

Reference Sample	Type of Variation	Number of True-Positive Variants	Number of False-Positive Variants	Number of False-Negative Variants	Sensitivity	Specificity	Mean Depth	VAF
S800-1	SNV/INDEL	27	0	0	100%	100%	1000×	2.00%
	CNV	3	0	0	100%	100%		2.00%
	Fusion	6	0	0	100%	100%		2.00%
S800-2	SNV/INDEL	27	0	0	100%	100%	2000×	0.50%
	CNV	3	0	0	100%	100%		0.50%
	Fusion	6	0	0	100%	100%		0.50%
Tru-Q 7	SNV/INDEL	39	0	1	97.50%	100%	993×	1–1.3%
Tru-Q 7	SNV/INDEL	37	0	4	93%	100%	560×	1–1.3%
mix of Tru-Q 7+ Tru-Q0 (wild type)	SNV/INDEL	34	0	0	85%	100%	794×	0.5–0.65%

**Table 3 ijms-26-05930-t003:** Comparison between the results of the OCA plus assay on FFPE (tissue findings, VAF) and the results of the 1021 assay (liquid-biopsy findings, Allele Frequency-AF) on the matched liquid-biopsy samples.

Patient	Tissue Findings Using the OCA Plus Assay	VAF	Liquid-Biopsy Findings with the 1021 Assay	AF
**Sample 1**	ERRFI1 c.1007_1016del	46%	ERRFI1 c.1007_1016del	0.50%
	TERT -146 T>C	35%	TERT -146 T>C	0.90%
	PALB2 c.1A>G	21%	PALB2 c.1A>G	0.80%
	TP53 c.743G>T	49.%	TP53 c.743G>T	1.60%
	CCND1 amplification	10 copies	CCND1 amplification	3.2 copies
	FGF19 amplification	10 copies	FGF19 amplification	3.2 copies
	FGF4 amplification	10 copies	FGF4 amplification	3.2 copies
	FGF3 amplification	10 copies	FGF3 amplification	3.2 copies
**Sample 2**	PIK3CA c.328_330del	7%	PIK3CA c.328_330del	4.90%
	CDH1 c.67C>T	10%	CDH1 c.67C>T	8.50%
**Sample 3**	EGFR c.2573T>G	34.%	EGFR c.2573T>G	32.40%
	TP53 c.413C>T	46%	TP53 c.413C>T	6.50%
**Sample 4**	KRAS c.38G>A	42%	Not Detected	
	BAP1 c.783+2T>C	22%	Not Detected	
	CDKN2A c.151-1G>T	29%	Not Detected	
**Sample 5**	NRAS c.183A>C	15%	Not Detected	
	APC c.4280delC	60%	Not Detected	
	ERBB3 c.695C>T	16%	Not Detected	
	KDMA c.641T>A	13.20%	Not Detected	
**Sample 6**	KRAS c.35G>A	28.60%	KRAS c.35G>A	3.00%
	RET c.2410G>A	32.44%	RET c.2410G>A	1.42%
	IDH1 c.395G>T	20.52%	IDH1 c.395G>T	0.98%
	TP53 c.614A>G	40.66%	TP53 c.614A>G	2.24%
**Sample 7**	EML4(6)–ALK(20)	2416 copies	EML4(6)–ALK(20)	1.20%
**Sample 8**	CTNNB1 c.121A>G	42.20%	Not detected	
	CD74-ROS1 fusion	3895 copies	Not detected	
**Sample 9**	TP53 c.814G>T	44.30%	TP53 c.814G>T	0.55%
**Sample 10**	KRAS c.35G>T	32.25%	KRAS c.35G>T	0.68%
	PIK3CA c.3139C>T	12.69%	PIK3CA c.3139C>T	1.72%
	MAP2K7 c.289C>T	13.47%	MAP2K7 c.289C>T	0.30%
	TP53 c.734G>A	11.70%	TP53 c.734G>A	0.60%

**Table 4 ijms-26-05930-t004:** Samples and variant types utilized in analytic validation.

Validated Variants	Validation Samples	Source of Validation
SNVs, Indels, CNVs, Fusions	50 clinical samples, 6 reference standards	FFPE/Tumor DNA
gLOH	37 clinical samples	FFPE/Tumor DNA
MSI	66 patient samples orthogonally tested for MSI status	FFPE
TMB	59 patient samples orthogonally tested for TMB status, 2 reference standards	FFPE/Tumor DNA

## Data Availability

Data is contained within the article or Appendix A: The original contributions presented in this study are included in the article/Appendix A. Further inquiries can be directed to the corresponding author(s).

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
