# Peer review of "Comprehensive Evaluation of a 1021-Gene Panel in FFPE and Liquid Biopsy for Analytical and Clinical Use"

_ijms, 2025, doi:10.3390/ijms26135930_

Round 1
Reviewer 1 Report
Comments and Suggestions for Authors
The article deals with the validation and evaluation of a commercial gene panel for biomarker analysis in solid tumors.
It is not clear to me whether any of the authors played any part in the development of the gene panel, or whether they are just providing data about the utility of a product they use in their routine laboratory analyses.
The analytical validation appears to be appropriate, well conducted, and adequately described.
On the other hand, despite the relevant number of patients tested, the results of clinical validation are incompletely presented and could be more detailed: for example, only percentages of the most frequently affected genes were provided, completely disregarding the rest, and the same approach is replicated for the other aspects considered. Since a large cohort of patients was studied, it would be of more interest to provide further information about the results, using detailed tables (which might be provided as supplementary files), followed by extensive discussion.
Furthermore, I would have found more interesting a comparison of the results in large cohorts of patients between this panel and other commercially available panels. Established that I would not expect that such a large number of samples can be tested with two or more different panels, I would suggest a comparison (and a thorough revision) with other similar studies that can be found in scientific literature, then highlighting the possible strengths, weaknesses, and novelty elements.
Minor recommendations:
- The commercial origin and name of the gene panel should be clearly stated in the introduction, since the whole paper is based on it.
- Appropriate details about ALL the materials and reagents mentioned should be supplied, such as the name of the manufacturer and the company's location (city, state, and country). This is also true for reference samples and standard used, for which also a specific reference (i.e.: citation) should be provided.
- The acronym OCA is never clearly defined.
Author Response
We would like to express our gratitude to the reviewer for his/her comments and the fruitful contribution to the improvement of the submitted manuscript. Please find below the responses to your comments:
1. It is not clear to me whether any of the authors played any part in the development of the gene panel, or whether they are just providing data about the utility of a product they use in their routine laboratory analyses.
We thank the reviewer for this important question.
The gene panel used in this study is a commercially available, ready-to-use assay. None of the authors were involved in its original design or development. Our study focused on validating the assay’s performance and evaluating its clinical utility in a routine diagnostic setting. This validation was carried out independently within a European clinical laboratory, where the assay had not previously been tested.
The analytical validation appears to be appropriate, well conducted, and adequately described.
- On the other hand, despite the relevant number of patients tested, the results of clinical validation are incompletely presented and could be more detailed: for example, only percentages of the most frequently affected genes were provided, completely disregarding the rest, and the same approach is replicated for the other aspects considered. Since a large cohort of patients was studied, it would be of more interest to provide further information about the results, using detailed tables (which might be provided as supplementary files), followed by extensive discussion.
To address this comment, we added actionable gene frequencies in pages 7-8 and supplementary material and figure “The actionability of variants was highly associated with the tumor type. Therefore, variants associated with on-label treatment were detected in 37.27% of breast cancer patients, 25.64% of prostate cancer patients, 24.35% of lung cancer patients, and 21.13% of the patients with biliary tract tumors. On the other hand, tumors such as cervical, endometrial, pancreatic cancer and sarcoma exhibited low frequencies of variants associated with on-label treatment (0%, 0%, 1.61% and 0.89%, respectively) (supplementary figure/file). In addition to on-label biomarkers several off label biomarkers are also detected in our cohort. PTEN alterations are particularly common in various tumor types including endometrial (36.67%), brain (30.47%), cervical (23.81%), thyroid (13.63%), sarcoma (10%), HNSCC (7.69%), underscoring its broad relevance in tumorigenesis (1). PIK3CA alterations are also frequently detected across cancer types, including ovarian (17.14%), urothelial (11.54%) and colorectal cancer (11.43%), reflecting its known role in PI3K pathway dysregulation(2). Moreover, KRAS alterations drive oncogenesis in the majority of pancreatic cancers (73.40%), while a high proportion of biliary tract tumors (25.35%) presents such alterations(3,4). Given their central role in tumorigenesis across diverse cancer types, such frequently mutated genes represent promising candidates for pan-cancer therapeutic targeting.
- Furthermore, I would have found more interesting a comparison of the results in large cohorts of patients between this panel and other commercially available panels. Established that I would not expect that such a large number of samples can be tested with two or more different panels, I would suggest a comparison (and a thorough revision) with other similar studies that can be found in scientific literature, then highlighting the possible strengths, weaknesses, and novelty elements.
Thank you for this comment.
We conducted a thorough research of the bibliography and provided further information concerning the utility/applicability of our panel compared to alternative assays in the discussion section (page 12)
“In our validation study, a direct comparison between the results obtained using the hybrid capture–based 1021-gene panel and the amplicon-based OCA Plus assay was performed in 50 clinical samples (5,6). The results showed key performance differences between the assays. The hybrid capture design enabled broader and more consistent interrogation of genomic regions of 1021 genes, including difficult-to-amplify or GC-rich loci. The new assay also achieved superior coverage uniformity, aided by the incorporation of unique molecular identifiers (UMIs) for error correction and accurate low-frequency variant detection, which can make it eligible for liquid biopsy analysis in addition to FFPE tissue testing (7). It is important to note that OCA Plus failed to detect a clinically relevant RAD51C variant in the genomic region due to insufficient coverage, which highlights a common limitation of amplicon-based approaches. These findings are in line with the results of other studies that compared hybrid capture and amplicon-based panels, which demonstrated that hybrid capture was more effective in detecting complex variants and fusions (8,9). Nevertheless, a notable advantage of OCA Plus and similar amplicon-based assays is their ability to operate with lower input DNA quantities, making them suitable for small or degraded samples. Moreover, while OCA+ offers practical advantages—such as low DNA input and rapid turnaround—it has limited sensitivity for subtle Copy Number Alterations, particularly deletions (10). In fact, a high concordance of 90.91%.was observed for ERBB2 amplification detection when comparing our hybrid capture methodology with the orthogonal assay (FISH) . Overall, while both methods have clinical utility, our results support the broader diagnostic reach and technical robustness of hybrid capture–based profiling, particularly in comprehensive genomic applications.
The 1021-panel employed in this study, showed comparable clinical utility to that of other hybrid capture-based assays, while its relevance in daily practice is further enhanced by the large panel design and the number of HR genes that were interrogated (8,11–16). Additionally, the DNA-only sequencing methodology was able to accurately identify gene fusions. This method was dependent on optimal probe design and extensive coverage to identify rearrangements in clinically pertinent fusion partners. In the absence of RNA-based analysis, the assay demonstrated robust performance in the detection of fusions. DNA and RNA have been employed in several CGP assays to enhance the detection sensitivity of fusions, particularly for genes such as ALK, RET, ROS1, and NTRK (17). However, RNA sequencing has also substantial practical and technical pitfalls, particularly when working with formalin-fixed, paraffin-embedded (FFPE) samples, where RNA integrity is frequently compromised. Therefore, although RNA-based methods can directly validate transcript expression and additionally identify rare or novel fusions, they require dual extraction procedures and high-quality input materials, leading to increased tissue waste, turn-around time, and complexity. Our findings support the clinical value of DNA-based, high-depth sequencing as a means for confident fusion detection, especially in the general diagnostic context where quality or even any RNA might be missing (16,18).”
Minor recommendations:
- The commercial origin and name of the gene panel should be clearly stated in the introduction, since the whole paper is based on it. Corrected
- Appropriate details about ALL the materials and reagents mentioned should be supplied, such as the name of the manufacturer and the company's location (city, state, and country). This is also true for reference samples and standard used, for which also a specific reference (i.e.: citation) should be provided.
Corrected
- The acronym OCA is never clearly defined. Corrected
Reviewer 2 Report
Comments and Suggestions for Authors
Dear Authors,
First of all, congratulations for your interesting work. I hope that my hints will help you in the next steps of improvement and the final manuscript will be really valuable for the readers. The abstract is exceptionally well-written and summarises the paper very well, congratulations on this - it is not easy to make such a good wrap-up.
There are several punctation mistakes (such as double space, double dot or no at all) and some typos - even if they do not change the value of the manuscript, I'd like to urge you to correct these imperfections.
It might be a good idea to explain the limitations of the liquid biopsy too, in a way that may concinve unconvinced people to use it more often. We know today, that this technology has been significantly improved, but it is not known by many people.
Also, maybe I've missed this information somehow, I'm sorry if I did, but have you performed targeted sequencing of only selected coding-regions of selected genes? Or entire gene sequence have been analysed?
Finally, I would like to thank you for the excellent figures and graphs you have prepared for the document, they enhance the value of your work and facilitate the understanding process.
Author Response
We would like to express our gratitude for the well-intentioned comments and clarifications of this reviewer. We are really pleased for his/her suggestions and we kindly provide you our responses:
- There are several punctation mistakes (such as double space, double dot or no at all) and some typos - even if they do not change the value of the manuscript, I'd like to urge you to correct these imperfections. Corrected
- It might be a good idea to explain the limitations of the liquid biopsy too, in a way that may convince unconvinced people to use it more often. We know today, that this technology has been significantly improved, but it is not known by many people.
Thank you for this comment.
Although a comprehensive validation of the assay in ctDNA was not the focus of this manuscript, we plan to present an in-depth validation in a separate publication using matched FFPE and plasma samples from the same patients. We have added the following statement to the Discussion section: “Liquid biopsy NGS analysis while important for biomarker identification, still presents several challenges. The main limitation is the lower sensitivity in early stage cancer and in patients undergoing cytotoxic or targeted therapy, since ctDNA levels may be extremely low, leading to false-negative results. Therapy can also suppress ctDNA shedding, rendering difficult to assess tumor mutational burden or minimal residual disease precisely during treatment. Furthermore, the detection of some genomic changes, such as gene fusions or copy number variations remains technically challenging with ctDNA technologies, though when used alongside tissue testing, it offers a more comprehensive molecular profile. [35]. However, in the present study 10 cases with tissue molecular profile results available underwent plasma NGS analysis. In 70% of these cases, the tissue alterations were also detectable in liquid biopsy, indicating the importance of utilizing this NGS procedure, despite its limitations. The findings identified in these cases were point mutations, CNVs and fusions, highlighting the applicability of this 1021-gene panel in liquid biopsies, as well. The acquisition of negative ctDNA findings in three cases may have been influenced by the fact that patients were undergoing treatment. Thus, the minimal quantity of ctDNA detected necessitates a subsequent analysis following treatment completion in order to ascertain whether the amount of ctDNA detected has increased. Moreover, the applicability of this assay in ctDNA analysis enhances its clinical utility, particularly in cases where tissue biopsy is not available. While the two methods are complementary in providing a comprehensive understanding of tumor biology, liquid biopsy can also be used for real-time monitoring of resistance mutations emerging after targeted therapy. Liquid biopsy molecular profiling has the potential to increase the number of patients eligible for targeted therapy therefore, it is recommended by international guidelines (NCCN, ESMO) for patient’s appropriate management[23].”
- Also, maybe I've missed this information somehow, I'm sorry if I did, but have you performed targeted sequencing of only selected coding-regions of selected genes? Or entire gene sequence have been analysed?
In the methodology section it is stated that: “The analysis included the entire exon regions of 312 genes, introns/promoters/fusion breakpoint regions of 38 genes and selected/partial coding exons of 709 genes (Table S1).”